# INVERSE CONSTRAINED REINFORCEMENT LEARNING

## ABSTRACT

Standard reinforcement learning (RL) algorithms train agents to maximize given reward functions. However, many real-world applications of RL require agents to also satisfy certain constraints which may, for example, be motivated by safety concerns. Constrained RL algorithms approach this problem by training agents to maximize given reward functions while respecting *explicitly* defined constraints. However, in many cases, manually designing accurate constraints is a challenging task. In this work, given a reward function and a set of demonstrations from an expert that maximizes this reward function while respecting *unknown* constraints, we propose a framework to learn the most likely constraints that the expert respects. We then train agents to maximize the given reward function subject to the learned constraints. Previous works in this regard have either mainly been restricted to tabular settings or specific types of constraints or assume knowledge of transition dynamics of the environment. In contrast, we empirically show that our framework is able to learn arbitrary *Markovian* constraints in high-dimensions in a model-free setting.

## 1 INTRODUCTION

Reward functions are a critical component in reinforcement learning settings. As such, it is important that reward functions are designed accurately and are well-aligned with the intentions of the human designer. This is known as agent (or value) alignment (see, e.g., Leike et al. (2018; 2017); Amodei et al. (2016)). Misspecified rewards can lead to unwanted and unsafe situations (see, e.g, Amodei & Clark (2016)). However, designing accurate reward functions remains a challenging task. Human designers, for example, tend to prefer simple reward functions that agree well with their intuition and are easily interpretable. For example, a human designer might choose a reward function that encourages an RL agent driving a car to minimize its traveling time to a certain destination. Clearly, such a reward function makes sense in the case of a human driver since inter-human communication is contextualized within a framework of unwritten and unspoken constraints, often colloquially termed as 'common-sense'. That is, while a human driver will try to minimize their traveling time, they will be careful not to break traffic rules, take actions that endanger passersby, and so on. However, we cannot assume such behaviors from RL agents since they are are not imbued with common-sense constraints.

Constrained reinforcement learning provides a natural framework for maximizing a reward function subject to some constraints (we refer the reader to Ray et al. (2019) for a brief overview of the field). However, in many cases, these constraints are hard to specify explicitly in the form of mathematical functions. One way to address this issue is to automatically extract constraints by observing the behavior of a constraint-abiding agent. Consider, for example, the cartoon in Figure 1. Agents start at the bottom-left corner and are rewarded according to how quickly they reach the goal at the bottom-right corner. However, what this reward scheme misses out is that in the real world the lower bridge is occupied by a lion which attacks any agents attempting to pass through it. Therefore, agents that are naïvely trained to maximize the reward function will end up performing poorly in the real world. If, on the other hand, the agent had observed that the expert (in Figure 1(a)) actually performed suboptimally with respect to the stipulated reward scheme by taking a longer route to the goal, it could have concluded that (for some unknown reason) the lower bridge must be avoided and consequently would have not been eaten by the lion!

Scobee & Sastry (2020) formalizes this intuition by casting the problem of recovering constraints in the maximum entropy framework for inverse RL (IRL) (Ziebart et al., 2008) and proposes a greedy

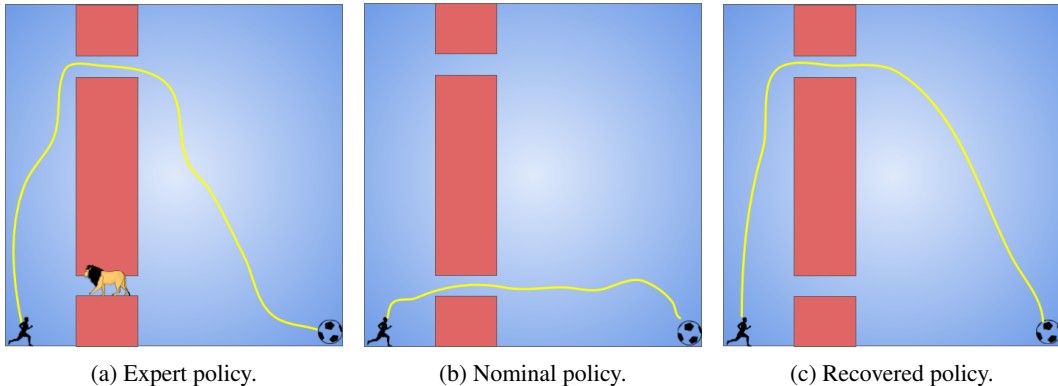

| (a) Expert policy. | (b) Nominal policy. | (c) Recovered policy. |

Figure 1: The TwoBridges environment. (a) The expert avoids the lion and takes the upper bridge. (b) Since the nominal policy is simply trained to get to the goal as quickly as possible, it instead takes the lower bridge. (c) Our method, on the other hand, is able to learn that the lower bridge should be avoided, and consequently our policy takes the upper bridge.

algorithm to infer the smallest number of constraints that best explain the expert behavior. However, Scobee & Sastry (2020) has two major limitations: it assumes (1) tabular (discrete) settings, and (2) the environment's transition dynamics. In this work, we aim to address both of these issues by *learning* a constraint function instead through a sample-based approximation of the objective function of Scobee & Sastry. Consequently, our approach is model-free, admits continuous states and actions and can learn arbitrary Markovian constraints[1]. Further, we empirically show that it scales well to high-dimensions.

Typical inverse RL methods only make use of expert demonstrations and do not assume any knowledge about the reward function at all. However, most reward functions can be expressed in the form "do this task while not doing these other things" where *other things* are generally constraints that a designer wants to impose on an RL agent. The main task ("do this") is often quite easy to encode in the form of a simple *nominal* reward function. In this work, we focus on learning the constraint part ("do not do that") from provided expert demonstrations and using it in conjunction with the nominal reward function to train RL agents. In this perspective, our work can be seen as a principled way to inculcate prior knowledge about the agent's task in IRL. This is a key advantage over other IRL methods which also often end up making assumptions about the agent's task in the form of regularizers such as in Finn et al. (2016).

The main contributions of our work are as follows:

- We formulate the problem of inferring constraints from a set of expert demonstrations as a *learning* problem which allows it to be used in continuous settings. To the best of our knowledge, this is the first work in this regard.
- We eliminate the need to assume, as Scobee & Sastry do, the environment's transition dynamics.
- We demonstrate the ability of our method to train constraint-abiding agents in high-dimensions and show that it can also be used to prevent reward hacking.

## 2 PRELIMINARIES

### 2.1 UNCONSTRAINED RL

A finite-horizon Markov Decision Process (MDP) $\mathcal{M}$ is a tuple $(\mathcal{S}, \mathcal{A}, p, r, \gamma, T)$, where $\mathcal{S} \in \mathbb{R}^{|\mathcal{S}|}$ is a set of states, $\mathcal{A} \in \mathbb{R}^{|\mathcal{A}|}$ is a set of actions, $p : \mathcal{S} \times \mathcal{A} \times \mathcal{S} \mapsto [0, 1]$ is the transition probability function (where $p(s'|s, a)$ denotes the probability of transitioning to state $s'$ from state $s$ by taking

---

[1]Markovian constraints are of the form $c(\tau) = \prod_{t=1}^{T} c(s_t, a_t)$ i.e. constraint function is independent of the past states and actions in the trajectory.

action $a$), $r : \mathcal{S} \times \mathcal{A} \mapsto \mathbb{R}$ is the reward function, $\gamma$ is the discount factor and $T$ is the time-horizon. A trajectory $\tau = \{s_1, a_1, \ldots, s_T, a_T\}$ denotes a sequence of states-action pairs such that $s_{t+1} \sim p(\cdot|s_t, a_t)$. A policy $\pi : \mathcal{S} \mapsto \mathcal{P}(\mathcal{A})$ is a map from states to probability distributions over actions, with $\pi(a|s)$ denoting the probability of taking action $a$ in state $s$. We will sometimes abuse notation to write $\pi(s, a)$ to mean the joint probability of visiting state $s$ and taking action $a$ under the policy $\pi$ and similarly $\pi(\tau)$ to mean the probability of the trajectory $\tau$ under the policy $\pi$.

Define $r(\tau) = \sum_{t=1}^{T} \gamma^t r(s_t, a_t)$ to be the total discounted reward of a trajectory. Forward RL algorithms try to find an optimal policy $\pi^*$ that maximizes the expected total discounted reward $J(\pi) = \mathbb{E}_{\tau \sim \pi}[r(\tau)]$. On the other hand, given a set of trajectories sampled from the optimal (also referred to as expert) policy $\pi^*$, inverse RL (IRL) algorithms aim to recover the reward function $r$, which can then be used to learn the optimal policy $\pi^*$ via some forward RL algorithm.

## 2.2 Constrained RL

While normal (unconstrained) RL tries to find a policy that maximizes $J(\pi)$, constrained RL instead focuses on finding a policy that maximizes $J(\pi)$ *while* respecting explicitly-defined constraints. A popular framework in this regard is the one presented in Altman (1999) which introduces the notion of a constrained MDP (CMDP). A CMDP $\mathcal{M}^c$ is a simple MDP augmented with a cost function $c : \mathcal{S} \times \mathcal{A} \mapsto \mathbb{R}$ and a budget $\alpha \geq 0$. Define $c(\tau) = \sum_{t=1}^{T} \gamma^t c(s_t, a_t)$ to be the total discounted cost of the trajectory $\tau$ and $J^c(\pi) = \mathbb{E}_{\tau \sim \pi}[c(\tau)]$ to be the expected total discounted cost. The forward constrained RL problem is to find the policy $\pi_c^*$ that maximizes $J(\pi)$ subject to $J^c(\pi) \leq \alpha$.

In this work, given a set $\mathcal{D}$ of trajectories sampled from $\pi_c^*$, the corresponding unconstrained MDP $\mathcal{M}$ (i.e., $\mathcal{M}^c$ without the cost function $c$) and a budget $\alpha$, we are interested in recovering *a* cost function which when augmented with $\mathcal{M}$ has an optimal policy that generates the same set of trajectories as in $\mathcal{D}$. We call this as the inverse constrained reinforcement learning (ICRL) problem.

If the budget $\alpha$ is strictly greater than $0$, then the cost function $c$ defines *soft* constraints over all possible state-action pairs. In other words, a policy is allowed to visit states and take actions that have non-zero costs as long as the expected total discounted cost remains less than $\alpha$. If, however, $\alpha$ is $0$ then the cost function translates into hard constraints over all state-action pairs that have a non-zero cost associated with them. A policy can thus never visit these state-action pairs. In this work, we restrict ourselves to this hard constraint setting. Note that this is not particularly restrictive since, for example, safety constraints are often hard constraints as well are constraints imposed by physical laws.

Since we restrict ourselves to hard constraints, we can rewrite the constrained RL problems as follows: define $\mathcal{C} = \{(s, a)|c(s, a) \neq 0\}$ to be the constraint set induced by $c$. The forward constraint RL problem is to find the optimal constrained policy $\pi_{\mathcal{C}}^*$ that maximizes $J(\pi)$ subject to $\pi_{\mathcal{C}}^*(s, a) = 0 \ \forall (s, a) \in \mathcal{C}$. The inverse constrained RL problem is to recover the constraint set $\mathcal{C}$ from trajectories sampled from $\pi_{\mathcal{C}}^*$.

Finally, we will refer to our unconstrained MDP as the nominal MDP hereinafter. The nominal MDP represents the nominal environment (simulator) in which we train our agent.

## 3 Formulation

### 3.1 Maximum Likelihood Constraint Inference

We take Scobee & Sastry as our starting point. Suppose that we have a set of trajectories $\mathcal{D} = \{\tau^{(i)}\}_{i=1}^{N}$ sampled from an expert $\pi_{\mathcal{C}}^*$ navigating in a constrained MDP $\mathcal{M}^{\mathcal{C}^*}$ where $\mathcal{C}^*$ denotes the (true) constraint set. Furthermore, we are also given the corresponding nominal MDP $\mathcal{M}$[2]. Our goal is to recover a constraint set which when augmented with $\mathcal{M}$ results in a CMDP that has an optimal policy that respects the same set of constraints as $\pi_{\mathcal{C}}^*$ does. Scobee & Sastry pose this as a maximum likelihood problem. That is, if we let $p_{\mathcal{M}}$ denote probabilities given that we are considering MDP $\mathcal{M}$ and assume a uniform prior on all constraint sets, then we can choose $\mathcal{C}^*$ according to

$$\mathcal{C}^* \leftarrow \arg\max_{\mathcal{C}} p_{\mathcal{M}}(\mathcal{D}|\mathcal{C}). \tag{1}$$

---

[2] Availability of transition dynamics model of nominal MDP is not necessary.

Under the maximum entropy (MaxEnt) model presented in Ziebart et al. (2008), the probability of a trajectory under a deterministic MDP $\mathcal{M}$ can be modelled as

$$\pi_{\mathcal{M}}(\tau) = \frac{\exp(\beta r(\tau))}{Z_{\mathcal{M}}} \mathbb{1}^{\mathcal{M}}(\tau), \tag{2}$$

where $Z_{\mathcal{M}} = \int \exp(\beta r(\tau)) \mathbb{1}^{\mathcal{M}}(\tau) d\tau$ is the partition function, $\beta \in [0, \infty)$ is a parameter describing how close the agent is to the optimal distribution (as $\beta \to \infty$ the agent becomes a perfect optimizer and as $\beta \to 0$ the agent simply takes random actions) and $\mathbb{1}$ is an indicator function that is 1 for trajectories feasible under the MDP $\mathcal{M}$ and 0 otherwise.

Assume that all trajectories in $\mathcal{D}$ are i.i.d. and sampled from the MaxEnt distribution. We have

$$p(\mathcal{D}|\mathcal{C}) = \frac{1}{(Z_{\mathcal{M}^{\mathcal{C}}})^N} \prod_{i=1}^{N} \exp(\beta r(\tau^{(i)})) \mathbb{1}^{\mathcal{M}^{\mathcal{C}}}(\tau^{(i)}). \tag{3}$$

Note that $\mathbb{1}^{\mathcal{M}^{\mathcal{C}}}(\tau^{(i)})$ is 0 for all trajectories that contain any state-action pair that belongs to $\mathcal{C}$. To maximize this, Scobee & Sastry propose a greedy strategy wherein they start with an empty constraint set and incrementally add state-action pairs that result in the maximal increase in $p(\mathcal{D}|\mathcal{C})$.

## 3.2 Sample-Based Approximation

Since we are interested in more realistic settings where the state and action spaces can be continuous, considering all possible state-action pairs individually usually becomes intractable. Instead, we propose a learning-based approach wherein we try to approximate $\mathbb{1}^{\mathcal{M}^{\mathcal{C}}}(\tau)$ using a neural network. Consider the loglikelihood

$$\mathcal{L}(\mathcal{C}) = \frac{1}{N} \log p(\mathcal{D}|\mathcal{C}) = \frac{1}{N} \sum_{i=1}^{N} \left[ \beta r(\tau^{(i)}) + \log \mathbb{1}^{\mathcal{M}^{\mathcal{C}}}(\tau^{(i)}) \right] - \log Z_{\mathcal{M}^{\mathcal{C}}},$$

$$= \frac{1}{N} \sum_{i=1}^{N} \left[ \beta r(\tau^{(i)}) + \log \mathbb{1}^{\mathcal{M}^{\mathcal{C}}}(\tau^{(i)}) \right] - \log \int \exp(\beta r(\tau)) \mathbb{1}^{\mathcal{M}^{\mathcal{C}}}(\tau) d\tau. \tag{4}$$

Note that $\mathbb{1}^{\mathcal{M}^{\mathcal{C}}}(\tau) = \prod_{t=0}^{T} \mathbb{1}^{\mathcal{M}^{\mathcal{C}}}(s_t, a_t)$ merely tells us whether the trajectory $\tau$ is feasible under the constraint set $\mathcal{C}$ or not. Let us have a binary classifier $\zeta_\theta$ parameterized with $\theta$ do this for us instead, i.e., we want $\zeta_\theta(s_t, a_t)$ to be 1 if $(s_t, a_t)$ is not in $\mathcal{C}^*$ and 0 otherwise. Using $\zeta_\theta(\tau)$ as a short hand for $\prod_{t=0}^{T} \zeta_\theta(s_t, a_t)$, we have

$$\mathcal{L}(\mathcal{C}) = \mathcal{L}(\theta) = \frac{1}{N} \sum_{i=1}^{N} \left[ \beta r(\tau^{(i)}) + \log \zeta_\theta(\tau^{(i)}) \right] - \log \int \exp(\beta r(\tau)) \zeta_\theta(\tau) d\tau. \tag{5}$$

Let $\mathcal{M}^{\bar{\zeta}_\theta}$ denote the MDP obtained after augmenting $\mathcal{M}$ with the cost function $\bar{\zeta}_\theta := 1 - \zeta_\theta$[3], and $\pi_{\mathcal{M}^{\bar{\zeta}_\theta}}$ denote the corresponding MaxEnt policy. Taking gradients of (5) with respect to $\theta$ gives us (see Appendix A.1 for derivation)

$$\nabla_\theta \mathcal{L}(\theta) = \frac{1}{N} \sum_{i=1}^{N} \nabla_\theta \log \zeta_\theta(\tau^{(i)}) - \mathbb{E}_{\tau \sim \pi_{\mathcal{M}^{\bar{\zeta}_\theta}}} \left[ \nabla_\theta \log \zeta_\theta(\tau) \right]. \tag{6}$$

Using a sample-based approximation for the right-hand term we can rewrite the gradient as

$$\nabla_\theta \mathcal{L}(\theta) \approx \frac{1}{N} \sum_{i=1}^{N} \nabla_\theta \log \zeta_\theta(\tau^{(i)}) - \frac{1}{M} \sum_{j=1}^{M} \nabla_\theta \log \zeta_\theta(\hat{\tau}^{(j)}), \tag{7}$$

where $\hat{\tau}$ are sampled from $\pi_{\mathcal{M}^{\bar{\zeta}_\theta}}$ (discussed in the next section). Notice that making $\nabla_\theta \mathcal{L}(\theta)$ zero essentially requires matching the expected gradient of $\log \zeta_\theta$ under the expert (left hand term) and

---

[3]Note that since we are assuming that $\alpha$ is 0, we can assign any non-zero (positive) cost to state-action pairs that we want to constrain. Here $1 - \zeta_\theta$ assigns a cost of 1 to all such pairs.

nominal (right hand term) trajectories. For brevity, we will write $\pi_{\mathcal{M}^{\bar{\zeta}_\theta}}$ as $\pi_\theta$ from now onwards. We can choose $\zeta_\theta$ to be a neural network with parameters $\theta$ and a sigmoid at the output. We train our neural network via gradient descent by using the expression for the gradient given above.

In practice, since we have a limited amount of data, $\zeta_\theta$, parameterized as a neural network, will tend to overfit. To mitigate this, we incorporate the following regularizer into our objective function.

$$R(\theta) = \delta \sum_{\tau \sim \{\mathcal{D}, \mathcal{S}\}} [\zeta_\theta(\tau) - 1] \qquad (8)$$

where $\mathcal{S}$ denotes the set of trajectories sampled from $\pi_\theta$ and $\delta \in [0, 1)$ is a fixed constant. $R$ incentivizes $\zeta_\theta$ to predict values close to 1, thus encouraging $\zeta_\theta$ to choose the smallest number of constraints that best explain the expert data.

### 3.3 FORWARD STEP

To evaluate (7) we need to sample from $\pi_\theta$. Recall that $\pi_\theta$ needs to maximize $J(\pi)$ subject to $\pi_\theta(s, a) = 0 \; \forall (s, a) \in \mathcal{Z}$ where $\mathcal{Z}$ is the constraint set induced by $\bar{\zeta}_\theta$. However, since $\zeta_\theta$ outputs continuous values in the range $(0, 1)$, we instead solve the *soft* constrained RL version, wherein we try to find a policy $\pi$ that maximizes $J(\pi)$ subject to $\mathbb{E}_{\tau \sim \pi}[\bar{\zeta}_\theta(\tau)] \leq \alpha$. In our experiments, we set $\alpha$ to a very small value. Note that if $\alpha$ is strictly set to 0 our optimization program will have an empty feasible set. Please refer to Appendix A.5 for some more discussion on $\alpha$.

We represent our policy as a neural network with parameters $\phi$ and train it by solving the following equivalent unconstrained min-max problem on the Lagrangian of our objective function

$$\min_{\lambda \geq 0} \max_{\phi} \mathcal{L}_F(\phi, \lambda) = J(\pi^\phi) + \frac{1}{\beta} \mathcal{H}(\pi^\phi) - \lambda(\mathbb{E}_{\tau \sim \pi^\phi}[\bar{\zeta}_\theta(\tau)] - \alpha) \qquad (9)$$

by gradient ascent on $\phi$ (via the Proximal Policy Optimization (PPO) algorithm (Schulman et al., 2017)) and gradient descent on the Lagrange multiplier $\lambda$. Note that we also add the entropy $\mathcal{H}(\pi^\phi) = -\mathbb{E}_{\tau \sim \pi^\phi}[\log \pi^\phi(\tau)]$ of $\pi^\phi$ to our objective function. Maximizing the entropy ensures that we recover the MaxEnt distribution in (2) at convergence (see Appendix A.3 for proof).

### 3.4 INCORPORATING IMPORTANCE SAMPLING

Running the forward step in each iteration is typically very time consuming. To mitigate this problem, instead of approximating the expectation in (6) with samples from $\pi_\theta$, we approximate it with samples from an older policy $\pi_{\bar{\theta}}$ where $\bar{\theta}$ were the parameters of $\zeta$ at some previous iteration. We, therefore, only need to learn a new policy after a fixed number of iterations. To correct for the bias introduced in the approximation because of using a different sampling distribution, we add important sampling weights to our expression for the gradient. In this case, the importance sampling weights can be shown to be given by (see Appendix A.2 for the derivation)

$$s(\tau) = \frac{\zeta_\theta(\tau)}{\zeta_{\bar{\theta}}(\tau)}. \qquad (10)$$

The gradient can thus be rewritten as

$$\nabla_\theta \mathcal{L}(\theta) \approx \frac{1}{N} \sum_{i=1}^{N} \nabla_\theta \log \zeta_\theta(\tau^{(i)}) - \frac{1}{\sum_{\hat{\tau} \sim \pi_{\bar{\theta}}} s(\hat{\tau})} \left[ \sum_{\hat{\tau} \sim \pi_{\bar{\theta}}} s(\hat{\tau}) \nabla_\theta \log \zeta_\theta(\hat{\tau}) \right]. \qquad (11)$$

Algorithm 1 summarizes our training procedure.

## 4 EXPERIMENTS

**Learning a single constraint:** Consider the TwoBrdiges environment in Figure 1. The agent starts at the bottom-left corner and can take one of the following 4 actions at each step: right, left, up, down. The reward in the left half is negative and in the right half is positive (and proportional to how close the agent is to the goal). This incentivizes the agent to cross over into the right half as

---

**Algorithm 1:** ICRL with Importance Sampling

---

**Input:** Expert trajectories $\mathcal{D}$, iterations $N$, number of backward iterations $B$.
Initialize $\theta$ and $\phi$ randomly.
**for** $i = 1, \ldots, N$ **do**
    Learn $\pi^\phi$ by solving (9) using current $\zeta_\theta$.
    **for** $j = 1, \ldots, B$ **do**
        Sample a set of trajectories $\mathcal{S} = \{\tau^{(k)}\}_{k=1}^M$ using $\pi^\phi$.
        Compute importance sampling weights $s(\tau^{(k)})$ using (10) for $k = 1, \ldots, M$.
        Use $\mathcal{S}$ and $\mathcal{D}$ to update $\theta$ via SGD by using the gradient in (11).
    **end**
**end**
**return** $\pi^\phi$

---

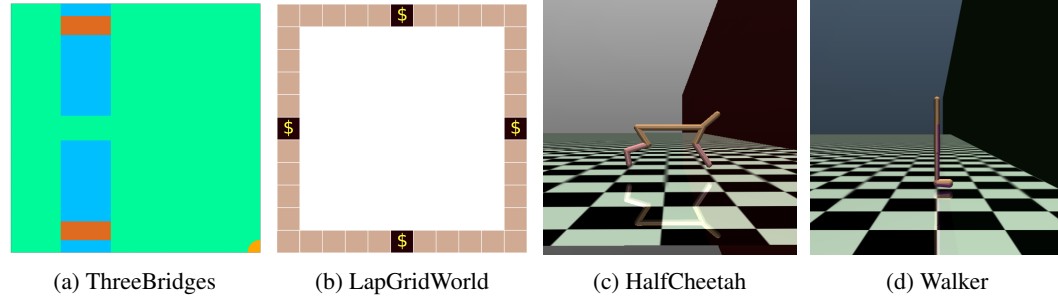

| (a) ThreeBridges | (b) LapGridWorld | (c) HalfCheetah | (d) Walker |

Figure 2: The environments used in the experiments. Note that nominal agents are not aware of the constraints shown.

quickly as possible, which is obviously via the lower bridge. However, since the lower bridge is occupied by a lion, the expert agent avoids it, whereas the nominal agent does not.

**Learning multiple constraints:** For this experiment we design the ThreeBridges environment shown in Figure 2(a). The agent starts at either the bottom or top-left corner with equal probability and, as in the TwoBridges case, is incentivized to cross over into the right half as quickly as possible. The expert on the other hand always takes the middle bridge, since both the upper and lower bridges are actually constrained.

**Preventing reward hacking:** Figure 2(b) shows the LapGridWorld environment[4] which we use to test if our method can prevent reward hacking. The agent is intended to sail clockwise around the track. Each time it drives onto a golden dollar tile, it gets a reward of 3. However, the nominal agent "cheats" by stepping back and forth on one dollar tile, rather than going around the track, and ends up getting more reward than the expert (which goes around the track, as intended).

**Scaling to high dimensions:** For this experiment, we use a simulated robot called HalfCheetah from OpenAI Gym (Brockman et al., 2016). The state and action spaces are of 18 and 6 dimensions respectively. The robot can move both in forward and backward directions and is rewarded proportionally to the distance it covers. For the constrained environment, shown in Figure 2(c), we add a solid wall at a distance of 5 units from the origin to prevent the robot from moving forwards. Consequently, the expert always moves backwards, whereas the nominal agent moves in both directions.

**Transferring constraints:** In many cases, constraints are actually part of the environment and are the same for different agents (for example, **all** vehicles must adhere to the same speed limit). In such instances, it is useful to first learn the constraints using only one agent and then transfer them onto other agents. As a proof of concept, we transfer the constraints learnt on the HalfCheetah agent from the previous paragraph on a Walker2d agent. Note that in this case $\zeta_\theta$ must only be fed a subset of the state and action spaces that are common across all agents. As such, we only train $\zeta_\theta$ on the

---

[4]This environment is similar to the boat race environment in Leike et al. (2017).

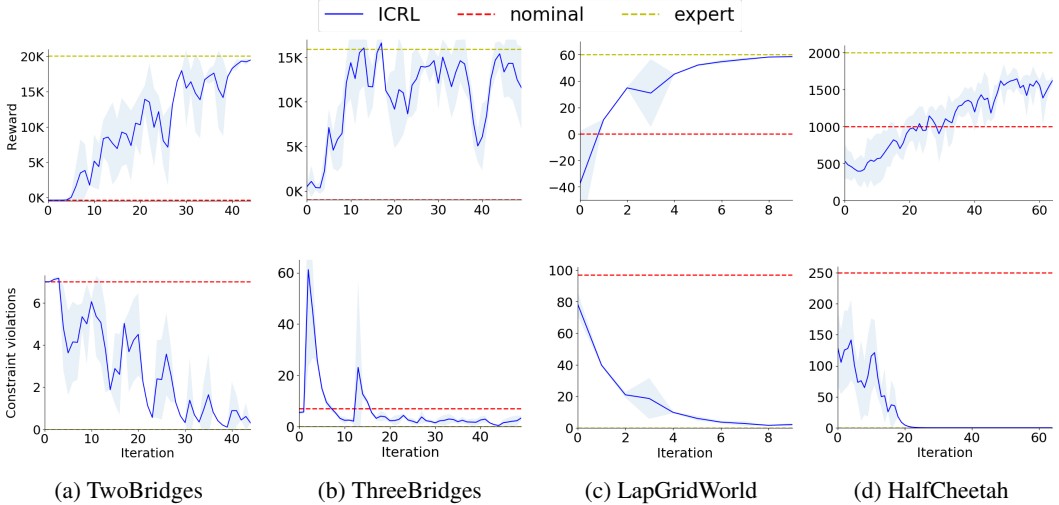

Figure 3: Performance of agents during training on their respective constrained (true) environments. All plots were smoothed and averaged over 3 seeds.

$(x, y)$ position coordinates of the HalfCheetah agent, since the rest of elements in the state space are specific to agent.

Figures 3 and 4 show the results of these experiments. The rewards shown are the actual rewards that the agent gets in the constrained (true) environment. In the case of TwoBridges and ThreeBridges, we add solid obstacles on the constrained bridges to prevent the agent from passing through them. In the constrained LapGridWorld environment, we award the agent 12 points whenever it completes a lap (rather than awarding 3 points each time it lands on a golden dollar tile, as in the nominal environment). Finally, in the case of HalfCheetah and Walker2d, we terminate the episode whenever the agent moves beyond the wall. The bottom row in Figure 3 shows the average number of constraints that the agent violates per trajectory when rolled out

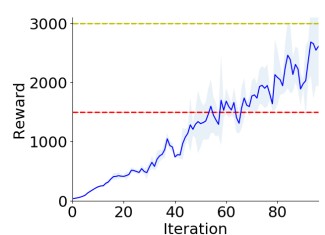

Figure 4: Walker2d

in the nominal environment. (For the LapGridWorld this is the number of times the agent attempts to move in the anti-clockwise direction.) For the Walker2d experiment, we observed $0$ constraint violations throughout training. This is because, in practice, $\zeta_\theta$ usually acts conservatively compared to the true constraint function (by also constraining state-action pairs that are close to the true constrained ones). Additional details on these experiments, including hyperparameters, can be found in Appendix A.4. As can be seen, over the course of training, the agent's true reward increases and its number of constraint violations go down.

## 5 ABLATION STUDIES

We conduct experiments on the TwoBridges environment to answer the following questions:

    (a) Can we learn constraints even when we have only *one* expert rollout?

    (b) Does importance sampling *speedup* convergence?

    (c) Does the regularization term in (8) encourage $\zeta_\theta$ to choose a *minimal* set of constraints?

To answer (a) we run our algorithm for different number of expert rollouts. As shown in Figure 5(a) we are able to achieve expert performance even with only one expert rollout.

For (b) we run experiments with and without importance sampling for different values of $B$ (the number of backward iterations). Figures 5(b) and 5(c) show the results. Using $B > 1$ without

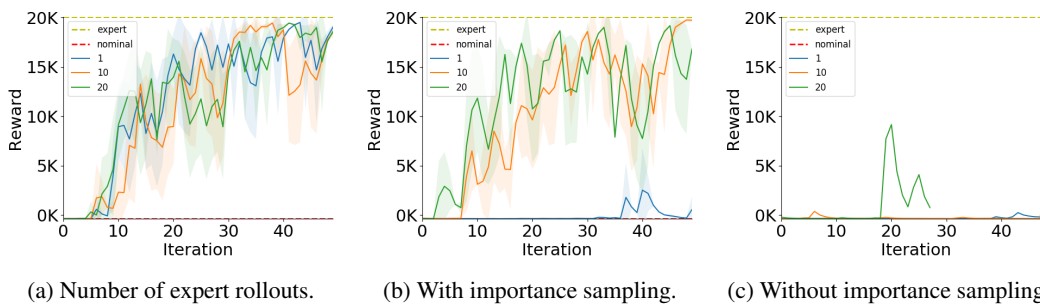

(a) Number of expert rollouts.    (b) With importance sampling.    (c) Without importance sampling.

Figure 5: Ablation study results. All plots were smoothed and (except in (c)) averaged over 3 seeds. Incomplete plots indicate a failure in the training procedure.

importance sampling results in a failure in the training procedure (usually resulting in NaNs) or low reward whereas using $B > 1$ with importance sampling accelerates convergence.

For (c) we run experiments with different values of $\delta$ which controls the extent of regularization (see (8)). Figure 6 shows the results. Note that as $\delta$ increases, $\zeta_\theta$ constrains fewer constraints. Also note that when $\delta = 1$, $\zeta_\theta$ fails to constrain any state.

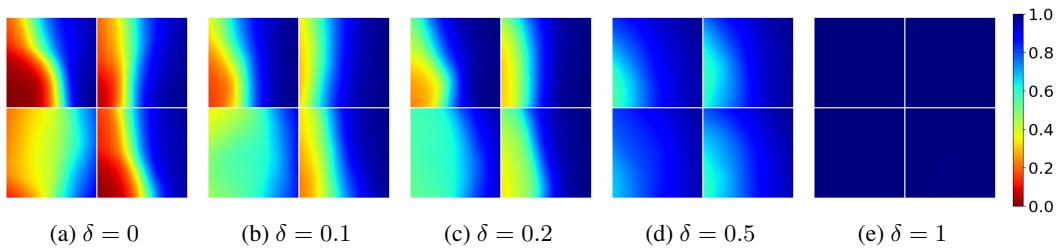

(a) $\delta = 0$      (b) $\delta = 0.1$      (c) $\delta = 0.2$      (d) $\delta = 0.5$      (e) $\delta = 1$

Figure 6: Heatmap of $\zeta_\theta$ for each of the four possible actions in TwoBridges (in clockwise direction from top-left): right, left, down, up.

## 6    RELATED WORK

**Forward Constrained RL:** Several approaches have been proposed in literature to solve the forward constrained RL problem in the context of CMDPs (Altman, 1999). Achiam et al. (2017) analytically solves trust region policy optimization problems at each policy update to enforce the constraints. Chow et al. (2018) uses a Lyapnuov approach and also provide theoretical guarantees. Le et al. (2019) proposes an algorithm for cases when there are multiple constraints. Finally, a well-known approach centers around rewriting the constrained RL problem as an equivalent unconstrained min-max problem by using Lagrange multipliers (Zhang et al., 2019; Tessler et al., 2019; Bhatnagar, 2010)) (see Section 3.3 for further details).

**Constraint Inference:** Previous work done on inferring constraints from expert demonstrations has either focussed on either inferring specific type of constraints such as geometric (D'Arpino & Shah, 2017; Subramani et al., 2018), sequential (Pardowitz et al., 2005) or convex (Miryoosefi et al., 2019) constraints or is restricted to tabular settings (Scobee & Sastry, 2020; Chou et al., 2018) or assume transition dynamics (Chou et al., 2020).

**Preference Learning:** Constraint inference also links to preference learning which aims to extract user preferences (constraints imposed by an expert on itself, in our case) from different sources such as ratings (Daniel et al., 2014), comparisions (Christiano et al., 2017; Sadigh et al., 2017), human reinforcement signals (MacGlashan et al., 2017) or the initial state of the agent's environment (Shah et al., 2019). Preference learning also includes inverse RL, which aims to recover an agent's reward function by using its trajectories. To deal with the inherent ill-posedness of this problem, inverse RL algorithms often incorporate regularizers (Ho & Ermon, 2016; Finn et al., 2016) or assume a prior

distribution over the reward function (Jeon et al., 2018; Michini & How, 2012; Ramachandran & Amir, 2007).

## 7    CONCLUSION AND FUTURE WORK

We have presented a method to learn constraints from an expert's demonstrations. Unlike previous works, our method both learns arbirtrary constraints and can be used in continuous settings.

While we consider our method to be an important first step towards learning arbitrary constraints in real-world continuous settings, there is still considerable room for improvement. For example, as is the case with Scobee & Sastry, our formulation is also based on (2) which only holds for deterministic MDPs. Secondly, we only consider hard constraints. Lastly, one very interesting extension of this method can be to learn constraints *only* from logs data in an offline way to facilitate safe-RL in settings where it is difficult to even build nominal simulators such as is the case for plant controllers.

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

# A APPENDIX

## A.1 GRADIENT OF LOG LIKELIHOOD

The gradient of (5) is

$$
\begin{aligned}
\nabla_\theta \mathcal{L}(\theta) &= \frac{1}{N} \sum_{i=1}^{M} \left[ 0 + \nabla_\theta \log \zeta_\theta(\tau^{(i)}) \right] - \frac{1}{\int \exp(\beta r(\tau)) \zeta_\theta(\tau) d\tau} \int \exp(\beta r(\tau)) \nabla_\theta \zeta_\theta(\tau) d\tau, \\
&= \frac{1}{N} \sum_{i=1}^{N} \nabla_\theta \log \zeta_\theta(\tau^{(i)}) - \int \frac{\exp(\beta r(\tau)) \zeta_\theta(\tau)}{\int \exp(\beta r(\tau')) \zeta_\theta(\tau') d\tau'} \nabla_\theta \log \zeta_\theta(\tau) d\tau, \\
&= \frac{1}{N} \sum_{i=1}^{N} \nabla_\theta \log \zeta_\theta(\tau^{(i)}) - \int p_{\mathcal{M}^{\bar{c}_\theta}}(\tau) \nabla_\theta \log \zeta_\theta(\tau) d\tau, \\
&= \frac{1}{N} \sum_{i=1}^{N} \nabla_\theta \log \zeta_\theta(\tau^{(i)}) - \mathbb{E}_{\tau \sim \pi_{\mathcal{M}^{\zeta_\theta}}} \left[ \nabla_\theta \log \zeta_\theta(\tau) \right],
\end{aligned}
\tag{12}
$$

where the second line follows from the identity $\nabla_\theta c_\theta(\tau) \equiv c_\theta(\tau) \nabla_\theta \log c_\theta(\tau)$ and the fourth line from the MaxEnt assumption.

## A.2 DERIVING THE IMPORTANCE SAMPLING WEIGHTS

Suppose that at some iteration of our training procedure we are interested in approximating the gradient of the log of the partition function $\nabla_\theta \log Z_\theta$ (where $\theta$ are the current parameters of our classifier) using an older policy $\pi_{\zeta_{\bar{\theta}}}$ (where $\bar{\theta}$ were the parameters of the classifier which induced the constraint set that this policy respects). We can do so by noting that

$$
\begin{aligned}
Z_\theta &= \int \exp(r(\tau)) \zeta_\theta(\tau) d\tau, \\
&= \int \pi_{\zeta_{\bar{\theta}}}(\tau) \left[ \frac{\exp(r(\tau)) c_\theta(\tau)}{\pi_{\zeta_{\bar{\theta}}}(\tau)} \right] d\tau, \\
&= \mathbb{E}_{\tau \sim \pi_{\zeta_{\bar{\theta}}}(\tau)} \left[ \frac{\exp(r(\tau)) c_\theta(\tau)}{\pi_{\zeta_{\bar{\theta}}}(\tau)} \right], \\
&= Z_{\bar{\theta}} \cdot \mathbb{E}_{\tau \sim \pi_{\zeta_{\bar{\theta}}}(\tau)} \left[ \frac{\zeta_\theta(\tau)}{\zeta_{\bar{\theta}}(\tau)} \right], \\
&\approx Z_{\bar{\theta}} \cdot \frac{1}{M} \sum_{\substack{i=1 \\ \tau \sim \pi_{\zeta_{\bar{\theta}}}}}^{M} \frac{\zeta_\theta(\tau^{(i)})}{\zeta_{\bar{\theta}}(\tau^{(i)})},
\end{aligned}
\tag{13}
$$

where the fourth lines follows from our MaxEnt assumption, i.e., $\pi_{\zeta_{\bar{\theta}}}(\tau) = \exp(r(\tau)) \zeta_{\bar{\theta}}(\tau) / Z_{\bar{\theta}}$.

Therefore

$$
\begin{aligned}
\nabla_\theta \log Z_\theta &= \frac{1}{Z_\theta} \nabla_\theta Z_\theta, \\
&= \frac{1}{Z_{\bar{\theta}} \cdot \frac{1}{M} \sum_{\tau \sim q(\tau)} \frac{\zeta_\theta(\tau)}{\zeta_{\bar{\theta}}(\tau)}} \left[ Z_{\bar{\theta}} \cdot \frac{1}{M} \sum_{\tau \sim q(\tau)} \frac{\nabla_\theta \zeta_\theta(\tau)}{\zeta_{\bar{\theta}}(\tau)} \right], \\
&= \frac{1}{\sum_{\tau \sim q(\tau)} \frac{\zeta_\theta(\tau)}{\zeta_{\bar{\theta}}(\tau)}} \left[ \sum_{\tau \sim q(\tau)} \frac{\zeta_\theta(\tau)}{\zeta_{\bar{\theta}}(\tau)} \nabla_\theta \log \zeta_\theta(\tau) \right].
\end{aligned}
\tag{14}
$$

### A.3 RATIONALE FOR (9)

Consider a constrained MDP $\mathcal{M}^{\mathcal{C}}$ as defined in Section 2.2. We are interested in recovering the following policy

$$\pi_{\mathcal{M}^c}(\tau) = \frac{\exp(\beta r(\tau))}{Z_{\mathcal{M}^c}} \mathbb{1}^{\mathcal{C}}(\tau), \tag{15}$$

where $Z_{\mathcal{M}^c} = \int \exp(\beta r(\tau)) \mathbb{1}^{\mathcal{C}}(\tau) d\tau$ is the partition function and $\mathbb{1}^{\mathcal{C}}$ is an indicator function that is 0 if $\tau \in \mathcal{C}$ and 1 otherwise.

**Lemma:** The Boltzmann policy $\pi^B(\tau) = \exp(\beta r(\tau))/Z$ maximizes $\mathcal{L}(\pi) = \mathbb{E}_{\tau \sim \pi}[r(\tau)] + \frac{1}{\beta}\mathcal{H}(\pi)$, where $H(\pi)$ denotes the entropy of $\pi$.

**Proof:** Note that the KL-divergence, $D_{KL}$, between a policy $\pi$ and $\pi^B$ can be written as

$$\begin{aligned}
\mathcal{D}_{KL}(\pi||\pi^B) &= \mathbb{E}_{\tau \sim \pi}[\log \pi(\tau) - \log \pi^B(\tau)], \\
&= \mathbb{E}_{\tau \sim \pi}[\log \pi(\tau) - \beta r(\tau) + \log Z], \\
&= -\mathbb{E}_{\tau \sim \pi}[\beta r(\tau)] - \mathcal{H}(\pi) + \log Z, \\
&= -\beta \mathcal{L}(\pi) + \log Z.
\end{aligned} \tag{16}$$

Since $\log Z$ is constant, minimizing $D_{KL}(\pi||\pi^B)$ is equivalent to maximizing $\mathcal{L}(\pi)$. Also, we know that $\mathcal{D}_{KL}(\pi||\pi^B)$ is minimized when $\pi = \pi^B$. Therefore, $\pi^B$ maximizes $\mathcal{L}$.

**Proposition:** The policy in (15) is a solution of

$$\underset{\lambda \geq 0}{\text{minimize}} \max_{\pi} \mathbb{E}_{\tau \sim \pi}[r(\tau)] + \frac{1}{\beta}\mathcal{H}(\pi^\phi) - \lambda(\mathbb{E}_{\tau \sim \pi^\phi}[\bar{\zeta}_\theta(\tau)] - \alpha). \tag{17}$$

**Proof:** Let us rewrite the inner optimization problem as

$$\max_{\pi} \mathbb{E}_{\tau \sim \pi}[r(\tau) - \lambda(\bar{\zeta}_\theta(\tau) - \alpha)] + \frac{1}{\beta}\mathcal{H}(\pi). \tag{18}$$

From the Lemma we know that the solution to this is

$$\pi(\tau, \lambda) = \frac{g(\tau, \lambda)}{\int g(\tau', \lambda) d\tau'}, \tag{19}$$

where $g(\tau, \lambda) = \exp(\beta(r(\tau) - \lambda(\bar{\zeta}_\theta(\tau) - \alpha)))$. To find $\pi^*(\tau) = \min_\lambda \pi(\tau, \lambda)$, note that:

1. When $\bar{\zeta}_\theta(\tau) \leq \alpha$, then $\lambda^* = 0$ minimizes $\pi$. In this case $g(\tau, \lambda^*) = \exp(\beta r(\tau))$.
2. When $\bar{\zeta}_\theta(\tau) > \alpha$, then $\lambda^* \to \infty$ minimizes $\pi$. In this case $g(\tau, \lambda^*) = 0$.

We can combine both of these cases by writing

$$\pi^*(\tau) = \frac{\exp(r(\tau))}{\int \exp(r(\tau')) \mathbb{1}^{\bar{\zeta}_\theta}(\tau') d\tau'} \mathbb{1}^{\bar{\zeta}_\theta}(\tau), \tag{20}$$

where $\mathbb{1}^{\bar{\zeta}_\theta}(\tau)$ is 1 if $\bar{\zeta}_\theta(\tau) \leq \alpha$ and 0 otherwise. (Note that the denominator is greater than 0 as long as we have at least one $\tau$ for which $\bar{\zeta}_\theta(\tau) \leq \alpha$, i.e., we have at least one feasible solution.) QED.

### A.4 EXPERIMENTAL SETTINGS

We used W&B (Biewald, 2020) to manage our experiments and conduct sweeps on hyperparameters. We used Adam (Kingma & Ba, 2015) to optimize all of our networks. All important hyperparameters are listed in Table 1. Details on the environments can be found below.

#### A.4.1 TWOBRIDGES

In this environment, the agent's state consists of its $(x, y)$ position coordinates. Agents start at $(0, 0)$ and the goal is at $(20, 0)$. Episdoes terminate when the agent is within one unit circle of the goal or when the number of timesteps exceeds 200. The bottom left corners of the bridges are at $(4, 5)$ and

$(4, 14)$. Each bridge is $4$ units long and $1$ unit wide. Agents can take one of the following actions: right, left, up and down. Each action moves the agent $0.7$ units in the respective direction. Agents attempting to move outside the $20 \times 20$ simulator or into the water (in between the bridges) end up in the same position and receive a reward of $-2$. The reward in the regions left of the bridge is fixed to $-1$ and on and to the right of the bridges is equal $10/d$ where $d$ is the Euclidean distance of the agent to the goal. Additionally, the agent's reward is scaled by $20$ if it is to the right of bridges and at or below the lower bridge (i.e. $y < 6$). Finally, the reward within one unit circle of the goal is fixed to $250$.

### A.4.2 THREEBRIDGES

This is similar to the ThreeBridges environment except that we now have three bridges. The bottom-left corners of each of the bridges are at $(4, 1)$, $(4, 9)$ and $(4, 17.5)$. The middle bridge is $4$ units long and $2$ units wide while the other two bridges are $4$ units long and $1.5$ units wide. Agents attempting to move outside the simulator or into water receive a reward of $-2$. The reward in regions left of the bridges is fixed to $-5$ and on and to the right of the bridges is $200/d$ where $d$ is Euclidean distance of the agent to the goal. The reward within one unit circle of the goal is fixed to $250$. Finally, agents randomly start at either the bottom-left of top-left corners with equal probability.

### A.4.3 LAPGRIDWORLD

Here, agents move on a $11 \times 11$ grid by taking either clockwise or anti-clockwise actions. The agent is awarded a reward $3$ each time it moves onto a bridge with a dollar (see Figure 2). The agent's state is the number of the grid it is on.

### A.4.4 HALFCHEETAH AND WALKER2D

The original reward schemes for HalfCheetah and Walker2d in OpenAI Gym (Brockman et al., 2016), reward the agents proportional to the distance they cover in the forward direction. We modify this and instead simply reward the agents according to the amount of distance they cover (irrespective of the direction they move in).

### A.5 ON THE BUDGET SIZE

As noted in Section 3.3, we set the budget size $\alpha$ to a very small value, typically around $0.01$. $\alpha$ controls the extent to which the agent respects the constraints imposed by $\zeta_\theta$. In this section, we study the effect of $\alpha$ on the agent's performance. We train an agent using our algorithm on the TwoBridges environment for different values of $\alpha$. Figure 7 shows the results. As can be seen, the agent's performance drops at higher values of $\alpha$. For example when $\alpha$ is $1$, the agent fails to achieve any meaningful reward.

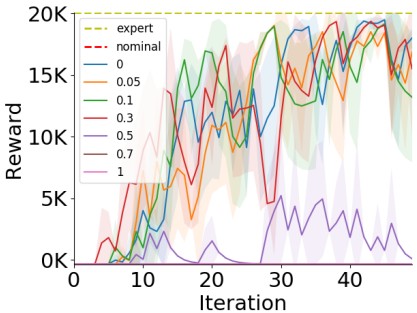

Figure 7: Performance of agent during training on the constrained (true) environment for different values for $\alpha$.

Table 1: Hyperparameters for various environments

| Parameter | TwoBridges | ThreeBridges | LapGridWorld | HalfCheetah |
|---|---|---|---|---|
| Rollout length | 200 | 200 | 200 | 500 |
| Batch size | 6000 | 10000 | 6000 | 20000 |
| Expert rollouts | 20 | 2 | 20 | 20 |
| Learning rate of policy | 1.23e-3 | 3.7e-3 | 1.23e-3 | 2.4e-2 |
| Learning rate of $\zeta_\theta$ | 0.001 | 0.005 | 0.001 | 0.001 |
| Learning rate of $\pi_\phi$ | 0.00124 | 0.00378 | 0.00124 | 0.0240 |
| Learning rate of $\lambda$ | 10 | 10 | 10 | 0.01 |
| Initial value of $\lambda$ | 1 | 1 | 1 | 1 |
| Budget, $\alpha$ | 0.01 | 0.01 | 0.01 | 0.01 |
| Entropy bonus, $1/\beta$ | 0.0871 | 0.1425 | 0.21 | 0.01 |
| GAE-$\gamma$ | 0.92 | 0.96 | 0.92 | 0.94 |
| GAE-$\lambda$ | 0.89 | 0.74 | 0.89 | 0.94 |
| Architecture of $\zeta_\theta$ | 64,64 | 64,64 | 64,64 | 10 |
| Architecture of $\pi_\phi$ | 64,64 | 64,64 | 64,64 | 64,64 |
| PPO clip ratio | 0.13 | 0.07 | 0.13 | 0.21 |
| PPO target kl | 0.01 | 0.01 | 0.01 | 0.01 |

