# OpenReview forum: "Inverse Constrained Reinforcement Learning"
_ICLR.cc/2021/Conference — Reject_

### Official Review · AnonReviewer2 · 2020-10-26
**Good algorithm derivation but empirical evaluation is lacking**

**Rating:** 4
**Confidence:** 4

**Review:**

This paper extends a tabular method for constraint inference to work in high-dimensional environments, and demonstrates it on a few environments.

Overall I liked the derivation of the algorithm (modulo some nitpicks described later). However, the experimental evaluation is sorely lacking: there are no baselines compared against, even though the obvious candidate to compare against is preference learning algorithms, e.g. imitation learning (such as GAIL [1]) and inverse reinforcement learning (such as AIRL [2]).

Indeed, it is not clear even theoretically what benefit constraint inference gives over the preference inference framework. While it is not common for preference inference to learn what _not_ to do (as this paper highlights), it is possible -- [3] is an example that creates some toy environments where the agent must learn what not to do, and then solves it using a single-state version of Maximum Causal Entropy Inverse Reinforcement Learning. I would expect that high-dimensional preference learning algorithms like GAIL and AIRL would also be able to do this, and thus are important to compare against.

In fact, it seems to me that the proposed algorithm can be cast as a special case of preference inference. This is most easily seen from the gradient expression in [4], where the MaxEnt IRL gradient in high-dimensional environments is written as

$$\frac{\delta r_{\theta}}{\delta \theta} [\mathbb{E} [ \mu ] - \mu_{D_{\tau}}].$$

This is the negative of what you might expect because they are defining a loss function to be minimized while you are defining a log likelihood to be maximized. (Note that $\mu$ is the state-occupancy measure.) If you set

$$r_{\theta}(\tau) = \log \zeta_{\theta}(\tau)$$

I believe you recover the proposed algorithm. So it seems that the main contribution relative to existing algorithms is to propose this particular structure on the reward function (which corresponds to a hard constraint because when $\zeta$ is zero the reward is negative infinity). So it becomes even more important to show why this particular structure is an improvement over e.g. the GAN-like structure in AIRL.

(Other minor contributions include the use of importance sampling and the regularization of $\zeta$, though it is also hard to tell how important these contributions are.)

Quality: As explained above, I have serious concerns about the experimental section.

Clarity: The paper was quite clear. I’ve listed a few minor cases below for improvement.

Originality: To my knowledge no other paper has inferred constraints based on the maximum entropy assumption in high-dimensional environments. However, there are both preference learning algorithms (discussed above) and algorithms that learn logic specifications [5], which seem similar in spirit to constraints (though are not the same).

Significance: Hard to judge without better experiments.

Nitpicks / typos:

> Note that this is not particularly restrictive since, for example, safety constraints are often hard constraints as well are constraints imposed by physical laws.

Shouldn’t physical laws (things like F = ma) already be encoded in the transition dynamics T? Why do we need constraints for this?

Equation (5) seems to have dropped a $\beta$ compared to Equation (4).

Should Equation (8) have an absolute value? Otherwise I don’t see how it incentivizes the classifier to predict values close to 1. Actually, I see, the network is constrained by the sigmoid to output values in [0, 1]. It could be worth clarifying this.

When going from Equation (5) to Equation (6), a policy π has suddenly appeared -- please explain what the policy is; the reader should not have to look at the appendix to understand the notation. (I assume it is a Boltzmann rational policy since you are using the MaxEnt framework.)

First line of Section 4: “TwoBrdiges” -> “TwoBridges”

References:

[1] Generative Adversarial Imitation Learning, https://arxiv.org/abs/1606.03476
[2] Learning Robust Rewards with Adversarial Inverse Reinforcement Learning, https://arxiv.org/abs/1710.11248
[3] Preferences Implicit in the State of the World, https://arxiv.org/abs/1902.04198
[4] Learning a Prior over Intent via Meta-Inverse Reinforcement Learning, https://arxiv.org/abs/1805.12573
[5] Maximum Causal Entropy Specification Inference from Demonstrations, https://arxiv.org/abs/1907.11792

---

> ### Author Response · Authors · 2020-11-18
> **Thank You For Your Response, We Address Your Comments Below**
>
> **Comment**: It is not clear even theoretically what benefit constraint inference gives over the preference inference framework. While it is not common for preference inference to learn what not to do (as this paper highlights), it is possible -- [3] is an example that creates some toy environments where the agent must learn what not to do, and then solves it using a single-state version of Maximum Causal Entropy Inverse Reinforcement Learning. I would expect that high-dimensional preference learning algorithms like GAIL and AIRL would also be able to do this, and thus are important to compare against.
>
> **Response**: In [general comment](https://openreview.net/forum?id=akgiLNAkC7P&noteId=OhR3UBnhlMz), we provide some discussion on _why_ GAIL and other popular IL methods should not be able to learn constraints effectively implicitly as part of shaped reward learning and empirically demonstrate the failure of GAIL on our TwoBridges environment.
>
> Further, as far as we understand, the setting of [3] (Rohin et al. 2019) is not that of standard IRL. Similar to the current work, they assume a misspecified reward function (akin to out nominal reward function) and their objective is also to _prevent_ negative actions on part of the agent or to _constrain_ the agent.
>
> Lastly, we note that GAIL is known to suffer from problems of poor generalization and domain shift (see section 4.1 and 4.4 of https://openreview.net/pdf?id=S1xKd24twB) to which our algorithm is obviously robust.
>
> **Comment**: In fact, it seems to me that the proposed algorithm can be cast as a special case of preference inference. This is most easily seen from the gradient expression in [4] ... So it seems that the main contribution relative to existing algorithms is to propose this particular structure on the reward function (which corresponds to a hard constraint because when $\zeta$ is zero the reward is negative infinity). So it becomes even more important to show why this particular structure is an improvement over e.g. the GAN-like structure in AIRL.
>
> **Response**: Please refer to our [general response](https://openreview.net/forum?id=akgiLNAkC7P&noteId=OhR3UBnhlMz); specifically point 1.
>
> **Comment**: Shouldn’t physical laws (things like F = ma) already be encoded in the transition dynamics T? Why do we need constraints for this?
>
> **Response**: One perspective on our work (and Scobee and Sastry) is to consider nominal (unconstrained) MDP as simulator and true (constrained) MDP as real world. In a real-world environment, physical laws are of course embedded in the transition model (and hence obeyed by expert), however, in simulators, there is a chance that dynamics may not be consistent with physical laws. An interesting example of this behavior can be seen in this [video](https://youtu.be/Lu56xVlZ40M?t=262). [1] uses a similar perspective for sim to real transfer.
>
> We have also fixed the typos in the updated version.
>
> [1] Off-Dynamics Reinforcement Learning: Training for Transfer with Domain Classifiers, arXiv preprint arXiv:2006.13916, 2020.

---

> > ### Comment · AnonReviewer2 · 2020-11-24
> > **Would still like clarification on the difference between IRL and ICRL**
> >
> > Apologies for the late reply.
> >
> > I hadn't properly considered the fact that ICRL and IRL have different use cases -- this is a fair point.
> >
> > > Please refer to our general response; specifically point 1.
> >
> > Point 1 of the general response says:
> >
> > > The main difference between IRL and ICRL is how they treat states that were not visited by the expert which contains both unconstrained (but low reward) states and constrained states. IRL algorithms treat all of these states equally and hence may violate constraints when the policy drifts away from states visited by expert. However, our method explicitly identifies the set of constrained states and ensures that they are avoided in all cases.
> >
> > I don't see how this is a response to the point I raised. To elaborate on my claim in more detail, I'm saying that if we take the neural net you use in your experiments, let's call it $f$, and then we add a log operation at the end of it to get $\log(f)$, and use that as the neural net reward model in Deep Maximum Entropy IRL (as formalized in [1]), then the resulting algorithm is _exactly identical_ to the algorithm proposed here. (I'm assuming here that the IRL algorithm is also provided the known part of the reward function that is available in the ICRL.) If the algorithms are exactly identical, there can't be any systematic differences in the behavior they produce. If you think there are systematic differences, then I must have made an error in my math somewhere; I'd be more convinced if you pointed out that error.
> >
> > On the specific points you mention, I don't see why IRL must treat unconstrained, low reward states equally to constrained states. It's perfectly possible for IRL to assign 0 reward to the unconstrained, low reward states, and -infinity (or -100) to the constrained states. Indeed, in the TwoBridges environment, I would typically expect that IRL will learn a lower reward on the close bridge than for other states in the environment.
> >
> > > Further, as far as we understand, the setting of [3] (Rohin et al. 2019) is not that of standard IRL.
> >
> > Yes, but the setting is solved using an algorithm based on IRL (specifically [2]). In any case this is a minor point and does not affect my score.
> >
> > Overall, there is more of a difference from IRL than I had thought. However, comparisons to IRL baselines still seem necessary, and I still think that the primary difference from existing deep IRL methods is the addition of a log to change the range to (-infinity, 0). So I've raised the score to a 4.
> >
> > [1] Learning a Prior over Intent via Meta-Inverse Reinforcement Learning, https://arxiv.org/abs/1805.12573
> >
> > [2] Ziebart et al., Modeling Interaction via the Principle of Maximum Causal Entropy, ICML 2010.

---

> > > ### Author Response · Authors · 2020-11-25
> > > **Further clarifications between IRL and ICRL**
> > >
> > > Thank you for revising your score.
> > >
> > > The main difference between IRL and ICRL is that ICRL handles constraints explicitly by assuming a reward function of the form $r(\tau) = r_{nominal}(\tau) + \log c(\tau)$ and only learns $c$ (here $c(\tau)$ is 0 if $\tau$ violates constraints and $1$ otherwise) whereas IRL tries to directly learn $r$. One major benefit of separating constraints is that they can then be transferred across agents that do not share the same nominal reward function. Furthermore, our expression for gradient is similar to the one in [1] since both of us start of from the same MaxEnt distribution. The addition of $\log$ in our paper is because we consider constraints separately, whereas [1] tries to learn a single reward function only.
> > >
> > > > On the specific points you mention, I don't see why IRL must treat unconstrained, low reward states equally to constrained states. It's perfectly possible for IRL to assign 0 reward to the unconstrained, low reward states, and -infinity (or -100) to the constrained states.
> > >
> > > Agreed, IRL _may_ distinguish between unconstrained, low reward states, but there is no reason to believe that it always _will_. Furthermore, algorithms like GAIL tend to suffer from distribution drift. This means that when the agent drifts away from expert distribution, it may start to act randomly, and in the process violate constraints. In fact, in some internal experiments, we observed that in the TwoBridges environment,  when we initialized an agent trained using ICRL from a different starting position (say, closer to the lower bridge), it achieved a high reward _without_ violating any constraints. Whereas agents trained using GAIL started acting randomly and violated constraints in the process.
> > >
> > >   \
> > > Finally, several prior works (e.g., [2,3]) consider the (forward) problem of maximizing a reward function subject to some constraints. ICRL should be thought of as the inverse of that problem (i.e., of recovering the constraints from a set of trajectories).
> > >
> > > ---
> > > [1] Xu et al. Learning a Prior over Intent via Meta-Inverse Reinforcement Learning, ICML 2019.  \
> > > [2] Achiam et al. Constrained Policy Optimization, ICML 2017.  \
> > > [3] Tessler et al. Reward Constrained Policy Optimization, ICLR 2019.

---

### Official Review · AnonReviewer1 · 2020-10-27
**Potentially interesting contribution, evaluation is the weakest aspect.**

**Rating:** 6
**Confidence:** 4

**Review:**

POST-REBUTTAL COMMENTS:
=========================

Several of the original concerns, mainly about the scope of the contribution, have been addressed by the rebuttal, so I've increased the overall score. See my responses to the rebuttal for the concerns that remain outstanding.

=========================


SUMMARY

This submission extends Scobee and Sastry's ICML-2020 work, which formulates the problem of learning (hard) constrains in RL as inverse constrained RL setting, by generalizing it from tabular to continuous settings and, empirically, increasing the latter's scalability.


HIGH-LEVEL REMARKS

I believe that the paper, while incremental, might be making a valuable contribution. At the same time, the magnitude of this contribution is difficult to assess, because paper makes a number of unsubstantiated claims about them, doesn't thoroughly analyze the limitations of its approach, doesn't provide an empirical comparison to existing alternatives, and has several other presentation issues as detailed below.


DETAILED COMMENTS

-- The paper claims, "[Scobee and Sastry] assume ability to modify the environment (specifically, to add arbitrary constraints to it)." I'm not sure what is meant here -- adding a constraint doesn't mean modifying an environment. The constraints apply to a policy.

-- The paper also claims, "our approach [...] does not suffer from the curse of dimensionality". This claim is unsubstantiated and doesn't sound true. Is the approach really oblivious to observation space dimensionality? Do you have a theorem showing this?

-- The problems in the experiments are too small to support the above claim about the approach being impervious to the curse of dimensionality empirically, if that was the intent. In particular, the paper doesn't experiment with environments involving visual observations.

-- How many expert trajectories were used in the experiments in Figures 3 and 4?

-- The paper claims that its approach can handle arbitrary constraints. This is a misleading claim, at best. There are rich temporal-logic based constraint languages such as LTL, CTL, etc. The paper doesn't discuss whether they are subsumed but CMDP's constraint form even theoretically. In the meantime, even the fragments of these constraint languages that can be compiled into CMDPs' constraints can cause exponential state space size explosion -- consider, e.g., constraints that specify that every trajectory must visit state A before state B before state C, etc. So in practice the paper's claim can't be true.

-- The paper's title, "CONSTRAINED REINFORCEMENT LEARNING WITH LEARNED CONSTRAINTS" isn't descriptive of the proposed technique. The paper assumes that the entire nominal MDP (the MDP without constraints) is given, along with a set of demonstration trajectories, and the agent tries to learn *only* the constraints. This means that once the constraints are learned, there is no more learning going on, only optimization. Thus, a more appropriate title for the paper would be something like "constraint learning via maxent inverse RL".

-- The paper's assumption of the nominal MDP being fully known, including the transition and reward function, is steep, stronger than IRL's. It also seems to require that the environment should be able to indicate constraint violations during training. All of this raises the question: why not solve the constrained MDP using one of existing forward methods for doing so (the paper mentions a few in the related work) without explicitly learning the constraints? Note that the paper's experiments don't just evaluate constraint learning, they evaluate solutions to the MDPs at hand, so a comparison to forward CMDP RL solvers is natural to ask for.

-- One answer to the above question can be that the goal of learning constraints is transferring them to other settings. But if so, the paper ought to analyze the effect of expert quality on the quality of constraints. Note that this effect can be drastic: if the expert is suboptimal, then its trajectories don't necessarily imply constraints, which means that the proposed methods might learn fictitious constraints that may make other environments where they are applied unsolvable.

-- In general, the paper doesn't compare its approach even to alternatives that also try to learn constraints explicitly. They may try to learn more constrained constraint types, but it would still be very useful to see a comparison to one of them (see the "constraint inference" part of the related work section).

-- The paper proposes to use a sigmoid at the output. I'm wondering whether this will lead to vanishing gradient for DNNs.

-- Related to the above point, note that while the paper claims to list all relevant hyperparameters for the experiments, it doesn't describe the network architecture (and the number of experts used in some of the experiments, as mentioned above) either in the main body or in the appendix, which makes its results unreproducible.

-- The use of J(.), with or without superscripts, for a value function is non-standard and confusing. The RL and controls convention is to use J(.) for a cost function to be minimized and V(.) for a value function to be maximized. The paper uses J both for a value function (the paragraph right before section 2.2) and, with a superscript, for a cost function (the first paragraph of 2.2). I strongly suggest bringing the paper in agreement with the standard convention, which will also remove the need to use J with a superscript.

-- In the related work is reasonably complete. A notable omission is Le, Voloshin, Yue, "Batch Policy Learning under Constraints", ICML-2019). Also, note that Tessler et al's and Subramani et al's papers' venue is listed in arXiv, but both have actually been published in refereed venues (ICLR-2019 for the former, CoRL-2018 for the latter). Please cite them as such.

-- Figure 5's legends are barely legible, please increase their font.

---

> ### Author Response · Authors · 2020-11-18
> **Thank You For Your Response, We Address Your Comments Below (2/2)**
>
> **Comment**: One answer to the above question can be that the goal of learning constraints is transferring them to other settings. But if so, the paper ought to analyze the effect of expert quality on the quality of constraints. Note that this effect can be drastic: if the expert is suboptimal, then its trajectories don't necessarily imply constraints, which means that the proposed methods might learn fictitious constraints that may make other environments where they are applied unsolvable.
>
> **Response**: Note that our formulation assumes that the expert follows a specific distribution (MaxEnt; equation 2). Note that the parameter $\beta$ controls the suboptimality of the expert, so our algorithm is already handling cases where the expert is suboptimal.
>
> **Comment**: In general, the paper doesn't compare its approach even to alternatives that also try to learn constraints explicitly. They may try to learn more constrained constraint types, but it would still be very useful to see a comparison to one of them (see the "constraint inference" part of the related work section).
>
> **Response**: We do not feel that a just comparison can be made to any of these works as these works assume knowledge that our approach does not. We have, however, added a note in Appendix A.6 elaborating shortcomings of traditional Inverse Reinforcement Learning (IRL).
>
> **Comment**: The paper proposes to use a sigmoid at the output. I'm wondering whether this will lead to vanishing gradient for DNNs.
>
> **Response**: In practice, the networks that we use are very small (typically 2 fully connected layers with 64 neurons each), that we do not face the problem of vanishing gradients.
>
> **Comment**: Related to the above point, note that while the paper claims to list all relevant hyperparameters for the experiments, it doesn't describe the network architecture (and the number of experts used in some of the experiments, as mentioned above) either in the main body or in the appendix, which makes its results unreproducible.
>
> **Response**: We've added the remainder of hyperparameters.
>
> **Comment**: The use of J(.), with or without superscripts, for a value function is non-standard and confusing. The RL and controls convention is to use J(.) for a cost function to be minimized and V(.) for a value function to be maximized. The paper uses J both for a value function (the paragraph right before section 2.2) and, with a superscript, for a cost function (the first paragraph of 2.2). I strongly suggest bringing the paper in agreement with the standard convention, which will also remove the need to use J with a superscript.
>
> **Response**: Our notation follows other papers in constrained RL [1,2]. J denotes the expected discounted reward (which needs to be maximized) and J^c denotes the expected discounted cost (which needs to be kept below a certain budget).
>
> **Comment**: In the related work is reasonably complete. A notable omission is Le, Voloshin, Yue, "Batch Policy Learning under Constraints", ICML-2019). Also, note that Tessler et al's and Subramani et al's papers' venue is listed in arXiv, but both have actually been published in refereed venues (ICLR-2019 for the former, CoRL-2018 for the latter). Please cite them as such.
>
> **Response**: We've added this citation, and have also expanded the Related Works section further.
>
> [1] Achiam et al. Constrained Policy Optimization, ICML 2017.
>
> [2] Tessler et al. Reward Constrained Policy Optimization, ICLR 2019.

---

> > ### Comment · AnonReviewer1 · 2020-11-19
> > **Thanks for your response. I'm increasing the score, but have a few remaining concerns.**
> >
> > Thank you, these responses address several of my concerns, so I'm increasing the score. The outstanding concerns are:
> >
> > *Rebuttal response*:  We do not assume transition dynamics.
> >
> > *My response to this*: This is at odds with what you say in the paper. The 3rd line of Section 3.1 says, "Furthermore, we are also given the corresponding nominal MDP M". Since you aren't excluding the transitition function, it stands to reason that you assume access to it. If you don't, it needs to be stated explicitly.
> >
> >
> > *Rebuttal response*:  We do not require the environment to indicate constraint violations [...], since the whole point of this work is to learn constraints.
> >
> > *My response to this*: This is akin to saying about supervised learning that you don't need labels since the whole point is learning a classifier regressor. In other words, I have trouble making sense both  of this response and, if you really don't need the environment to tell you when your learned policy violates a constraint, of how you actually learn the constraints. You need to get a training signal that helps your agent improve its knowledge of constraints as it is learning, don't you?
> >
> >
> > *Rebuttal response*: Our notation follows other papers in constrained RL [1,2]
> >
> > *My response to this*: This is a minor point, but these papers are by no means an authority on MDP notation, and there are lots of other papers with even more confusing notation -- just because they were published doesn't mean they are worth imitating. The convention established in multiple textbooks is to use J for cost-based value functions and V for reward-based value functions. Doing otherwise is likely to cause confusion.
> >
> >
> >
> > *One part of rebuttal response*: We are currently working on an experiment involving visual observations and are fairly confident that our approach will be able to work in this setting as well. Note that prior works on IRL (e.g, [4,5])) have not shown any difficulty in scaling to visual observations.
> >
> > *Another part of rebuttal response, to my remark about the use of sigmoid function being a perf. bottleneck*: In practice, the networks that we use are very small (typically 2 fully connected layers with 64 neurons each), that we do not face the problem of vanishing gradients.
> >
> > *My response to these*: these comments are at odds. Perhaps IRL hasn't had issues with visual observations, but IIRC prior IRL work isn't using signmoids with DNNs. Once you add a conv. net to handle visual observations and thereby make the network deeper, it's far from obvious that sigmoid won't become an issue.
> >
> >
> >
> > *Rebuttal response*: We do not feel that a just comparison can be made to any of these [constraint inference] works.
> >
> > *My response to this*: I really don't see why not. E.g., why can't you can evaluate against Scobee & Sastry on Two Bridges and LapGridWorld, both of which are discrete-space and hence can be solved in a tabular fashion.
> >
> > Even if a comparison to any single piece of prior work across all benchmarks is impossible, without a comparison to *some* prior works on *some* benchmarks it's difficult to assess this method.
> >
> >
> > Empirical evaluation is currently the paper's biggest weakness, I would say.

---

> > > ### Author Response · Authors · 2020-11-20
> > > **Thank You For Revising Your Score**
> > >
> > > **Response By Reviewer**: This is at odds with what you say in the paper. The 3rd line of Section 3.1 says, "Furthermore, we are also given the corresponding nominal MDP M". Since you aren't excluding the transitition function, it stands to reason that you assume access to it. If you don't, it needs to be stated explicitly.
> > >
> > > **Response 2 By Authors**: We have added a footnote at that point clarifying this.
> > >
> > >
> > > **Response By Reviewer**: This is akin to saying about supervised learning ... You need to get a training signal that helps your agent improve its knowledge of constraints as it is learning, don't you?
> > >
> > > **Response 2 By Authors:**: The key philosophy of this work (and Scobee & Sastry) is that if nominal agent (using only nominal reward) does something which was not done by expert then that was because expert is aware of some constraints which nominal agent is not. We try to identify the most likely constraints that expert is aware of based on the discrepancy between the behaviours of expert and nominal. In a very loose sense, you can think of expert trajectories as having label 1 and nominal trajectories as having label 0.
> > >
> > >
> > > **Response By Reviewer**: these comments are at odds. Perhaps IRL hasn't had issues with visual observations, but IIRC prior IRL work isn't using signmoids with DNNs. Once you add a conv. net to handle visual observations and thereby make the network deeper, it's far from obvious that sigmoid won't become an issue.
> > >
> > > **Response 2 By Authors:**: Sigmoid is only necessary at the last layer of our neural network. Internal layers can use any activation function (we use ReLU). As far as we know, this setting is quite general in discriminator in GAN based/inspired works and works reasonably well.
> > >
> > >
> > > **Response By Reviewer**: Even if a comparison to any single piece of prior work across all benchmarks is impossible, without a comparison to some prior works on some benchmarks it's difficult to assess this method.
> > >
> > > **Response 2 By Authors**:  We agree and are working on rectifying this.

---

> > > > ### Comment · AnonReviewer1 · 2020-11-22
> > > > **OK, sounds good.**
> > > >
> > > > --

---

> ### Author Response · Authors · 2020-11-18
> **Thank You For Your Response, We Address Your Comments Below (1/2)**
>
> You are requested to please see the [general comment](https://openreview.net/forum?id=akgiLNAkC7P&noteId=OhR3UBnhlMz) as well. We address your detailed comments turn by turn below.
>
> **Comment**: The paper claims, "[Scobee and Sastry] assume ability to modify the environment (specifically, to add arbitrary constraints to it)." I'm not sure what is meant here -- adding a constraint doesn't mean modifying an environment. The constraints apply to a policy.
>
> **Response**: We misspoke. What we meant was that Scobee & Sastry assume transition dynamics. We have clarified this is in the paper.
>
> **Comment**: The paper also claims, "our approach [...] does not suffer from the curse of dimensionality". This claim is unsubstantiated and doesn't sound true. Is the approach really oblivious to observation space dimensionality? Do you have a theorem showing this? The problems in the experiments are too small to support the above claim about the approach being impervious to the curse of dimensionality empirically, if that was the intent. In particular, the paper doesn't experiment with environments involving visual observations.
>
> **Response**: Previous approaches (such as Scobee and Sastry) are only applicable in tabular settings. Hence, these approaches explicitly require enumeration of all states and actions. Therefore, their space complexity is exponential in the number of dimensions i.e. $\mathcal{O}(k^n)$ where $n$ is the total number of dimensions of state and action space. Our approach, on the other hand, uses (deep) neural network based function approximation to learn a parametric representation of constraint set (as well as policy) which has been empirically shown to scale better [1,2,3]. We have clarified the sentence "our appraoch [..] ..." to reflect this thinking.
>
> In addition, we are currently working on an experiment involving visual observations and are fairly confident that our approach will be able to work in this setting as well. Note that prior works on IRL (e.g, [4,5])) have not shown any difficulty in scaling to visual observations.
>
> **Comment**: How many expert trajectories were used in the experiments in Figures 3 and 4?
>
> **Response**: We've added this to the list of hyperparameters in A.5.
>
> **Comment**: The paper claims that its approach can handle arbitrary constraints. This is a misleading claim, at best. There are rich temporal-logic based constraint languages such as LTL, CTL, etc. The paper doesn't discuss whether they are subsumed but CMDP's constraint form even theoretically. In the meantime, even the fragments of these constraint languages that can be compiled into CMDPs' constraints can cause exponential state space size explosion -- consider, e.g., constraints that specify that every trajectory must visit state A before state B before state C, etc. So in practice the paper's claim can't be true.
>
> **Response**: The reviewer is correct that constraints with temporal dependence can not be _efficiently_ represented in CMDP framework and our approach inherits this deficiency. In general, our approach can learn arbitrary Markovian constraints as we are operating in the setting of MDPs; specifically _any_ constraint that can be encoded in RL policy can be learned. We have qualified our claim with 'Markovian' in the revised manuscript.
>
>
> **Comment**: The paper's title, "CONSTRAINED REINFORCEMENT LEARNING WITH LEARNED CONSTRAINTS" isn't descriptive of the proposed technique ... Thus, a more appropriate title for the paper would be something like "constraint learning via maxent inverse RL".
>
> **Answer**: Agreed. We've changed the title to "Inverse Constrained Reinforcement Learning".
>
> **Comment**: The paper's assumption of the nominal MDP being fully known, including the transition and reward function, is steep, stronger than IRL's ... so a comparison to forward CMDP RL solvers is natural to ask for.
>
> **Answer**: We **do not** assume transition dynamics. Also, we do not require the environment to indicate constraint violations (the constraint violations plotted in Figure 3 are just for reporting purposes; the algorithm does not require them), since the whole point of this work is to _learn_ constraints. We cannot compare against forward constraint learning algorithms since they require the constraint function to be known (which they aren't in our case). Expert, however, can be considered as proxy to the performance of forward constraint learning algorithms using true constraint function.
>
> [1] Deep Reinforcement Learning: A Brief Survey, IEEE Signal Processing Magazine,2017.
>
> [2] Learning Complex Dexterous Manipulation with Deep Reinforcement Learning and Demonstrations, Robotics: Science and Systems, 2018.
>
> [3] Stochastic Latent Actor-Critic: Deep Reinforcement Learning with a Latent Variable Model, NeurIPS 2020.
>
> [4] Guided Cost Learning, ICML, 2016.
>
> [5] Inverse reinforcement learning for video games, NeurIPS Deep RL Workshop, 2018.

---

### Official Review · AnonReviewer3 · 2020-10-28
**Constrained Reinforcement Learning with Learned Constraints**

**Rating:** 5
**Confidence:** 4

**Review:**

Summary of review:
Interesting work, clearly described and well executed. But weak support for the overall motivation, and incomplete baselines to confirm the approach is useful.

Description:
This paper describes a new approach for inverse constraint learning, whereby an RL agent infers constraints (on state/action pairs) from expert trajectories, and uses this in combination for a given reward function to optimize an agent’s behavior.

The main novelty in the work is the ability to infer constraints with fewer assumptions than previous (incl. handling continuous state/action spaces). The work is still limited to deterministic domains.

The paper is clearly written, well organized, easy to follow.  The authors are transparent about their assumptions throughout.  Interesting ablation studies are included in the experiments. The characterization of the results & discussion are fair and not over-inflated.

My main concerns with the work are: (1) the proposed setting itself, and (2) some missing baselines in the experiments.  Regarding the setting, I don’t really buy the motivation.  There are claims (p.2) that rewards are easier to express than constraints, and therefore it is reasonable to assume that the reward is given but the constraints are inferred. But I don’t see the support for this (experimental, theoretical or from the literature) either in people or in AI.  Furthermore, it is not obvious to me that this is a better strategy than straight-forward IRL.  Under some (reasonably flexible) assumptions, constraints can be transformed into (negative) rewards.  So why not just apply IRL to infer a reward function that includes both the positive (goal) and the negative (constraints)?  At the very least, this should be considered as a baseline in the experiments, applying one of the recent IRL methods, such as GAIL, to see whether there is any advantage to this formulation.  If anything, the proposed method makes a stronger assumption on knowledge (having the reward function) than most IRL approaches, so those should perform worse in practice.

Additional questions:
-	Sec.3.1: Is it necessary for D (the observed trajectories) to be drawn from \pi*_C?  Could the trajectories be drawn from other domains?  The results near the end on transfer suggest so.  Why not embrace this in the initial formulation?
-	Sec.3.1 / 3.2: Can you clarify how you use the given reward function?  I assume this is r(\tau) in Eqn 2, but if that’s the case, it may not be clear why you need to know the reward function a priori, rather than be able to observe it during trajectories.
-	Sec.3.2: How do you draw negative examples for your binary classifier?  It seems all the trajectory data would be considered as positive examples.
-	Sec.3.4:  I’m somewhat surprised that the importance sampling helps that much. In many other instances, the variance is too high to see an improvement in practice, especially when the importance weights are calculated over a full trajectory (not one-step).  Any idea why it works so well (as per Fig.5) in this setting?
-	Fig.3:  Is the x-axis the number of trajectories of constraint data (i.e. D)?   In Fig.3(b),  Why do you break the constraints here?  Is this indicative of not enough data?  Why don’t you show the constraints for Walk2d (Fig.4)?  I strong recommend some analysis of the results in the main text to make sure to explicitly state the findings, not leave it to the reader’s reading of the plots.
-	Why do you use only 3 seeds?  That seems insufficient, as per recent discussion on reproducibility in RL.  How were those chosen?
-	You briefly mention some directions for future work in the last paragraph of Sec.7.  Can you also comment on what is the difficulty in each?  Why is the current work limited to deterministic MDPs, or hard constraints, or on-policy trajectories?


Minor comments:
-	Intro: “This is known as agent alignment” -> It is also known as value alignment, and there are many more references.
-	If you need additional space, some (or all) of Fig.2 and Fig.6 could be moved to appendix.
-	The references need to be cleaned up. Stick to a consistent style (full first name, or initial, not a mix).  A few references don’t say “where” the work was published. A few references have the arxiv link, even though the work since has been accepted at a peer-reviewed venue.  This type of sloppiness gives a bad impression (suggesting the authors might be as sloppy with their scientific work.)

---

> ### Author Response · Authors · 2020-11-18
> **Thank You For Your Response, We Address Your Comments Below (2/2)**
>
> **Comment**: You briefly mention some directions for future work in the last paragraph of Sec.7. Can you also comment on what is the difficulty in each? Why is the current work limited to deterministic MDPs, or hard constraints, or on-policy trajectories?
>
> **Response**: The current work is limited to deterministic MDPs since equation 2, which was proposed in [1], only holds for deterministic MDPs. To deal with stochastic MDPs, we will need to reformulate the current work using the framework presented in [2] - which is an extension of [1] for stochastic MDPs. The current work is limited to hard constraints since we assume a budget of 0 (section 2.2). We are not exactly sure at the moment how to extend this work to soft constraints. Finally, learning from log data in the current framework is not feasible due to the fact that we need to sample from $\pi_{\mathcal{M}^{\bar{\zeta}_\theta}}$ for constraint function update in (7) and thus require access to the nominal environment for this purpose.
>
> [1] Ziebart et al., Maximum Entropy IRL, AAAI 2008.
> [2] Ziebart et al., Modeling Interaction via the Principle of Maximum Causal Entropy, ICML 2010.
>
> ## Response To Minor Comments
>
> We've addressed the minor comments in the updated manuscript.

---

> ### Author Response · Authors · 2020-11-18
> **Thank You For Your Response, We Address Your Comments Below (1/2)**
>
> ## Response To Major Comments
>
> We address the motivation for this work in the context of limitations of IRL in the [general comment](https://openreview.net/forum?id=akgiLNAkC7P&noteId=OhR3UBnhlMz). We also try to address the issue of missing baselines there. We address your additional questions as follows:
>
> **Comment**: Sec.3.1: Is it necessary for D (the observed trajectories) to be drawn from $\pi^*_C$? Could the trajectories be drawn from other domains? The results near the end on transfer suggest so. Why not embrace this in the initial formulation?
>
> **Response**: We believe that it is possible to have expert trajectories from a different domain, though we have not experimentally verified this yet. However, note that the reward functions in both domains must be the same.
>
> **Comment**: Sec.3.1 / 3.2: Can you clarify how you use the given reward function? I assume this is r(\tau) in Eqn 2, but if that’s the case, it may not be clear why you need to know the reward function a priori, rather than be able to observe it during trajectories.
>
> **Response**: Yes $r(\tau)$ is the _nominal_ reward function. It is required to solve the problem in (9) (page 5). Note that $J(\pi) = E[r(\tau)]$ and we optimize the objective in (9) using the standard 'sample and train' approach in on-policy RL.
>
> **Comment**: Sec.3.2: How do you draw negative examples for your binary classifier? It seems all the trajectory data would be considered as positive examples.
>
> **Response**: We are not really doing negative sampling here. This is what happens: we start out with an estimate of the constraint function $\zeta$. We then train a policy $\pi$ to produce trajectories that respect this particular choice of constraint function using equation 9. These trajectories are then used to update $\zeta$ according to equation 7. Then policy $\pi$ is updated to respect the new constraint function and the iterative procedure continues.
>
> **Comment**: Sec.3.4: I’m somewhat surprised that the importance sampling helps that much. In many other instances, the variance is too high to see an improvement in practice, especially when the importance weights are calculated over a full trajectory (not one-step). Any idea why it works so well (as per Fig.5) in this setting?
>
> **Response**: We believe importance sampling works well because constraints only evolve gradually and hence current trajectories are often not drastically different from previous iterations. However, in a few cases, where zeta did change rapidly, we observed that importance sampling did not work well (the importance sampling ratio either vanished or exploded). This can typically be fixed by decreasing the learning rate of zeta.
>
>
> **Comment**: Fig.3: Is the x-axis the number of trajectories of constraint data (i.e. D)? In Fig.3(b), Why do you break the constraints here? Is this indicative of not enough data? Why don’t you show the constraints for Walk2d (Fig.4)? I strong recommend some analysis of the results in the main text to make sure to explicitly state the findings, not leave it to the reader’s reading of the plots.
>
> **Response**: No, the x-axis represents the iteration number of the algorithm (`i' in Algorithm 1). We have added some analysis of the results in the main text.
> Regarding Walker2d, true cost in this environment is trivially zero (this is the same constraint function as in figure 3d) and hence its graph is just a straight line at 0. We have included this information in the main text as well.
>
> **Comment**: Why do you use only 3 seeds? That seems insufficient, as per recent discussion on reproducibility in RL. How were those chosen?
>
> **Response**: We will add results on 10 random seeds (these 3 seeds were also randomly chosen) during the next revision period. To help reproducibility, we also plan to release our code alongside camera-ready submission.

---

### Official Review · AnonReviewer4 · 2020-10-28
**Official Blind Review | Reviewer #4**

**Rating:** 7
**Confidence:** 4

**Review:**

#### Summary

The submission focuses on a variant of inverse reinforcement learning, where the learner knows the task reward but is unaware of hard constraints that need to be respected while completing the task. The authors provide an algorithm to recover these constraints from expert demonstrations. The proposed algorithm builds upon a recent technique (Scobee & Sastry 2020) and addresses problems with large and continuous state spaces.

++++++++++++++++++++++++++++++++++
#### Reasons for score

Strengths:
* The problem considered is interesting and relevant to the ICLR community.
* The technique developed (Algorithm 1) is novel and well motivated.
* The experiments provide adequate evidence to back the claims.
* The paper is very well written and organized.

Weaknesses:
* Justification for the policy loss function (Equation 9) is unclear.
* Comparison with prior art is lacking.
* Discussion of related work is sparse and can be more detailed.

Based on the above-mentioned strengths, I vote for accepting. My concerns (further detailed below) potentially can be addressed during the rebuttal phase.

++++++++++++++++++++++++++++++++++
#### Major Comments

1. (page 2) The requirement of ‘ability to modify the environment’ is listed as a limitation of prior art (Scobee & Sastry 2020). However, like the current approach, the prior art adds the constraints / modifies the environments only conceptually (and not physically). Further, both the current and prior work focus on the case of hard constraints. Please clarify this limitation of the prior art vis-à-vis proposed approach.

2. (page 2) The rationale behind the objective (Equation 7) of the prior art and the proposed approach is identical. Please clarify, then, if the current algorithm is also greedy.

3. (Equation 9) Please provide additional details for the inclusion of the entropy term in the policy loss function.
  - The principle of maximum entropy is used to arrive at Eq. 4, the loss function of theta (since Eq. 4 uses the term derived in Eq. 2, which in turn is obtained from the maximum entropy principle). Given this, it is unclear why the entropy term is also included in Eq. 9. Is it used as a regularizer?
  - Alternatively put, consider the unconstrained version of Equation 9. In this unconstrained case, the problem is analogous to MaxEnt IRL (Ziebart et al.). In MaxEnt IRL, given the reward $\theta$, the policy $\phi$ is computed by value / policy iteration and without the extra entropy term.
  - Further, adding both $J$ and $H$ in the loss seem counterintuitive as they have different ‘units’. J is cumulative reward, while H is dimensionless entropy. Why is the entropy term normalized by $\beta$? How is the normalization constant chosen?

4. (Section 4) While not all domains considered in the Experiments can be captured by the prior art (Scobee & Sastry 2020), the first three can be (as they have discrete state, action spaces). Please benchmark the proposed approach with prior art for these three domains. Time permitting, also consider utilizing one of the recent high-dimensional techniques (see below) as another baseline.

5. (Section 6) Space permitting, please include a discussion of following related works.

  - Constrained IRL for high-dimensional problems:

    * Chou, Glen, Necmiye Ozay, and Dmitry Berenson. "Learning parametric constraints in high dimensions from demonstrations." Conference on Robot Learning. PMLR, 2020.

    * Park, Daehyung, et al. "Inferring Task Goals and Constraints using Bayesian Nonparametric Inverse Reinforcement Learning." Conference on Robot Learning. PMLR, 2020. Notes: Extends beyond the proposed approach to consider constraints which may not be global (i.e., locally active constraints).

    * Chou, Glen, Necmiye Ozay, and Dmitry Berenson. "Learning constraints from locally-optimal demonstrations under cost function uncertainty." IEEE Robotics and Automation Letters 5.2 (2020): 3682-3690.

  - Inverse reward / policy learning frameworks that incorporate prior knowledge of reward / policy:

    * Ramachandran, Deepak, and Eyal Amir. "Bayesian Inverse Reinforcement Learning." IJCAI. Vol. 7. 2007.

    * Michini, Bernard, and Jonathan P. How. "Bayesian nonparametric inverse reinforcement learning." Joint European conference on machine learning and knowledge discovery in databases. Springer, Berlin, Heidelberg, 2012.

    * Unhelkar, Vaibhav V., and Julie A. Shah. "Learning models of sequential decision-making with partial specification of agent behavior." Proceedings of the AAAI Conference on Artificial Intelligence. Vol. 33. 2019.

    * Jeon, Wonseok, Seokin Seo, and Kee-Eung Kim. "A bayesian approach to generative adversarial imitation learning." Advances in Neural Information Processing Systems. 2018.

  - Learning features (which can be in the form of logical constraints) for IRL:

    * Choi, Jaedeug, and Kee-Eung Kim. "Bayesian nonparametric feature construction for inverse reinforcement learning." Twenty-Third International Joint Conference on Artificial Intelligence. 2013.

++++++++++++++++++++++++++++++++++
#### Questions for Rebuttal Phase

Please address comments 1-4.

++++++++++++++++++++++++++++++++++
#### Minor Comments

- (typo) In the Introduction, Scobee & Sastry is used as singular noun, where in fact it is plural.

- (Equation 5) Beta is missing in the log exponential term.

- (Page 4, below Equation 7) The statement ‘Notice that … essentially tries to match’ is ambiguous, since the gradient by itself does not try to match the two values. Please consider rephrasing to say that this matching occurs at the minima (where the gradient is zero).

- (Page 5, Section 3.3) Please denote the range $[0,1]$ as $(0,1)$, since 0 and 1 are not in the range of $\zeta$.

- (Equation 9) Consider distinguishing the loss functions in Equation 5 and 9 (say through superscript or subscript). Due to $L$ being overloaded, at first glance, I misunderstood the loss function in Eq 9 as a continued derivation of Eq 5.

- (Section 7, typo) (2) -> Eq. (2)

++++++++++++++++++++++++++++++++++

---

> ### Author Response · Authors · 2020-11-18
> **Thank you for your review, we address your comments below**
>
> ## Response to major comments
>
> **Comment**: (page 2) The requirement of ‘ability to modify the environment’ is listed as a limitation of prior art (Scobee & Sastry 2020). However, like the current approach, the prior art adds the constraints / modifies the environments only conceptually (and not physically). Further, both the current and prior work focus on the case of hard constraints. Please clarify this limitation of the prior art vis-à-vis proposed approach.
>
> **Response**: We misspoke. What we meant was that Scobee & Sastry assume transition dynamics, while our approach relaxes this assumption. We have clarified this in the updated version of the paper.
>
> **Comment**: (page 2) The rationale behind the objective (Equation 7) of the prior art and the proposed approach is identical. Please clarify, then, if the current algorithm is also greedy.
>
> **Response**: Our algorithm simply chooses the most likely constraint set (akin to Scobee & Sastry). However, while Scobee & Sastry solve this maximum likelihood problem via a greedy approach, we instead use gradient descent on the objective function. Of course, gradient descent can be considered as a greedy algorithm (which greedily converges to a local minima).
>
> **Comment**: (Equation 9) Please provide additional details for the inclusion of the entropy term in the policy loss function.
>
> **Response**: We've added a section on the rationale behind (9) in the appendix (A.3). In short: solving (9) yields the maximum entropy policy (equation 2).
>
> **Comment**: (Section 4) While not all domains considered in the Experiments can be captured by the prior art (Scobee & Sastry 2020), the first three can be (as they have discrete state, action spaces). Please benchmark the proposed approach with prior art for these three domains. Time permitting, also consider utilizing one of the recent high-dimensional techniques (see below) as another baseline.
>
> **Response**: Scobee & Sastry assume access to transition dynamics and hence there is no fair way to compare with their work. As other reviewers had suggested, we have directed our efforts towards building a case against IRL (see the [general comment by authors](https://openreview.net/forum?id=akgiLNAkC7P&noteId=OhR3UBnhlMz)). We are currently working through the references on high dimensional works that you have provided and considering if we can setup a fair comparison with any of them.
>
> **Comment**: (Section 6) Space permitting, please include a discussion of following related works.
>
> **Response**: We've expanded our Related Works section by discussing some of the works you have mentioned.
>
> ## Response to minor comments
> Thank you for pointing out these errors; they have all been attended to.

---

> > ### Comment · AnonReviewer4 · 2020-11-24
> > **Thank you for the detailed response and updates to the manuscript.**
> >
> > Except for comparison with prior work, most of my comments have been address.
> >
> > Based on the ongoing discussion, please further clarify / discuss
> > - (Comment 4) How does the approach compare to the baselines suggested in the reviews? The responses provide a qualitative discussion for some of the baselines. However, as pointed across the reviews, a quantitative comparison is necessary to assess the method.
> > - (related to the comments by AnonReviewer 1) Please clarify the signal (in addition to the expert demonstrations), if any, the learner needs from the environment regarding constraint violations.
> > - Include a discussion on false positives (i.e., constraints learned when none exist) of the approach.

---

> > > ### Author Response · Authors · 2020-11-25
> > > **Response By Authors**
> > >
> > > > (related to the comments by AnonReviewer 1) Please clarify the signal (in addition to the expert demonstrations), if any, the learner needs from the environment regarding constraint violations.
> > >
> > > Environment in CIRL does not provide any constraint violation signal. Environment only provides reward, next observation and termination (done) signals. We then use _learned_ constraint function to get the constraint violation signal for updating nominal policy via eq (9). We then improve on our estimate of constraint function by using samples from nominal policy and expert policy via eq (7). This process continues iteratively.
> > > Possibly, this confusion emanated from our GAIL experiment; we have elaborated on the setup for that experiment in [response to Reviewer 2](https://openreview.net/forum?id=akgiLNAkC7P&noteId=lMS_w2j7bt1) comment on our general comment.
> > >
> > > > Include a discussion on false positives (i.e., constraints learned when none exist) of the approach.
> > >
> > > False positives are possible. This is why we incorporate the regularizer in Eq (8) to penalize the network if it tries to add constraints needlessly.

---

### Author Response · Authors · 2020-11-18
**General Comment By Authors | Why Inverse Constrained Reinforcement Learning (ICRL) and not Inverse Reinforcement Learning (IRL)?**

In this comment, we provide a general rebuttal regarding why preference learning frameworks and IRL are not well suited to the situations where expert behavior is moderated by cost functions in addition to reward function.

1. **ICRL and IRL promote different behaviors away from expert**
The main difference between IRL and ICRL is how they treat states that were not visited by the expert which contains both unconstrained (but low reward) states and constrained states. IRL algorithms treat all of these states equally and hence may violate constraints when the policy drifts away from states visited by expert. However, our method explicitly identifies the set of constrained states and ensures that they are avoided in _all_ cases.

2. **ICRL and IRL Differ In Use Cases**
Standard IRL algorithms (GAIL, Guided Cost Learning) assume that you have no information about reward whatsoever; however, ICRL assumes that (nominal) reward is known. This makes the use cases of ICRL and IRL quite distinct.

3. **IRL and IL algorithms 'smoothness' assumption is in direct conflict with discontinuous behavior of shaped reward**
The only way to truly subsume constraints into the reward function is by assigning a reward of negative infinity on the constrained states. This, however, will result in a discontinuous reward function. It is not immediately clear how IRL algorithms like GAIL and Guided Cost Learning will fare in this case, especially since they often incorporate regularizers which encourage smooth behaviors (e.g, by penalizing magnitudes of first or second-order gradients).
4. **Constraint Transfer**
When constraints are independent of the agent, then the cost function learned by using ICRL may be used to train new agents (with possibly very different nominal reward functions and morphologies).
5. **IRL and IL Algorithms may not mimic expert accurately when expert and training agent domains differ**
In addition, IRL and IL algorithms train their policies in the same domain as the expert. However, ICRL assumes that there are two domains: an "expert" or "constrained MDP" domain where the expert acts (and eventually the nominal will also act) and a "training" or "nominal MDP" domain in which the nominal agent is trained. We note that when training and expert domains differ, it is no longer guaranteed that the training agent will accurately mimic the expert behavior.
To demonstrate this behavior, we train an agent using GAIL on the _constrained_ and _nominal_ versions of our TwoBrdiges environment. Note that in this environment, there are two bridges, where the lower bridge is constrained in the _true_ version of the environment. Expert samples are collected in the constrained version of the environment (by blocking the lower bridge). When GAIL is trained on this (constrained) environment, it accurately learns to go through the top bridge, similar to the expert. However, when GAIL is trained on the _nominal_ version of the environment, it violates the constraints and goes through the lower bridge. We want to specifically emphasize that only difference between the two experiments was that of training environment; everything else was same.

The results can be seen in this [anonymous google drive folder](https://drive.google.com/drive/folders/1byLrfMl8DKujK_8AhhFlYuZik0igfGK8?usp=sharing).

[This figure](https://drive.google.com/file/d/104bsGgXmasFeaXMJk41PDlj9M9RO5bQ1/view?usp=sharing) shows expert data (generated in constrained environment) plotted on a random (gail) discriminator (dots denote states visited by the agent).

[This figure](https://drive.google.com/file/d/19zYSAef01r7LYRZJxyzfbyKuabZhX1ZX/view?usp=sharing) shows final policy samples from agent trained in nominal environment plotted on corresponding final (gail) discriminator.

[This figure](https://drive.google.com/file/d/1rdKaOYGUr9etDG7jC-uupdjWIMOGq2k6/view?usp=sharing) shows final policy samples from agent trained in constrained environment plotted on corresponding final (gail) discriminator.

---

> ### Comment · AnonReviewer2 · 2020-11-24
> **What does it mean to train GAIL on the nominal version of the environment?**
>
> Specifically, in response to:
>
> > However, when GAIL is trained on the nominal version of the environment, it violates the constraints and goes through the lower bridge.
>
> What exactly does this mean? My understanding is that for IRL / ICRL, the environment does not provide a "constraint violation" signal, so what's the difference between the nominal version and the constrained version of the environment?
>
> Perhaps you compared between GAIL trained on demonstrations collected from experts trained in the nominal environment with GAIL trained on demonstrations collected from experts trained in an environment with a constraint signal? Or are the demonstrations identical in both cases?

---

> > ### Author Response · Authors · 2020-11-25
> > **Clarification on enviroment/MDPs**
> >
> > You are correct; for ICRL/IRL, the environment does not provide a "constraint violation" signal. However, a CMDP $<S,A,P,R,C>$ with hard constraints can be converted to MDP by using dynamics shaping $<S,A,\hat P, R>$ by including the constraints in the dynamics model.  Specifically,
> >
> > $$\hat P(s'|s,a) = 0 \hspace{83pt} if \hspace{10pt} C(s,a,s') > 0$$
> > $$\hat P(s'|s,a) =  \frac{P(s'|s,a)}{\int_S \int_A P(s'|s,a) da ds} \hspace{26pt} if \hspace{10pt} C(s,a, s') = 0$$
> >
> > We convert _true_ CMDP to the corresponding _true_ MDP and train GAIL on this MDP. This is what we refer to as constrained environment in the original environment. In addition, we train GAIL on the nominal environment which is simply the MDP $<S, A, P, R>$. GAIL does not violate constraints on MDP $<S,A,\hat P, R>$ but does violate constraints on MDP $<S,A,P,R>$.
> >
> > Expert data in both cases from CMDP $<S,A,P,R,C>$. All other training variables are also the same across both experiments.
> >
> > Why this is so?
> > We do not have a definitive answer but we hypothesize that this is probably because the constrained region is very small in this case and states on either side of the constrained region are part of or very near to expert trajectories; hence, making it difficult for the discriminator to differentiate between them.

---

### Decision · Program_Chairs · 2021-01-07
**Final Decision**

**Decision:**

Reject

**Comment:**

This paper introduces ICRL, where the RL agent is supposed to maximize the reward under unknown constraints, which should be inferred from the expert demonstration. Reviewers generally agreed that this is an interesting work, and potentially make RL to be applied to more general settings. However, they also would like to see more experimental results with baselines (e.g. agents based on IRL and also related constrained learning approaches) to make the motivation behind the approach more convincing. I hope these concerns are addressed in the future work.